# Altered Spinal Homeostasis and Maladaptive Plasticity in GFAP Null Mice Following Peripheral Nerve Injury

**DOI:** 10.3390/cells11071224

**Published:** 2022-04-05

**Authors:** Ciro De Luca, Assunta Virtuoso, Sohaib Ali Korai, Raffaella Cirillo, Francesca Gargano, Michele Papa, Giovanni Cirillo

**Affiliations:** 1Neural Network Morphology & Systems Biology Lab, Division of Human Anatomy, Department of Mental and Physical Health and Preventive Medicine, University of Campania “Luigi Vanvitelli”, 80138 Naples, Italy; ciro.deluca@unicampania.it (C.D.L.); assunta.virtuoso@unicampania.it (A.V.); sohaib.ali@studenti.unicampania.it (S.A.K.); raffaella.cirillo@unicampania.it (R.C.); michele.papa@unicampania.it (M.P.); 2Unit of Anesthesia, Intensive Care and Pain Management, Department of Medicine, Campus Bio-Medico University of Rome, 00128 Rome, Italy; f.gargano@unicampus.it; 3SYSBIO Centre of Systems Biology ISBE.ITALY, University of Milano-Bicocca, 20126 Milano, Italy

**Keywords:** GFAP, reactive astrogliosis, peripheral nerve injury, spinal cord, neuropathic behavior

## Abstract

The maladaptive response of the central nervous system (CNS) following nerve injury is primarily linked to the activation of glial cells (reactive gliosis) that produce an inflammatory reaction and a wide cellular morpho-structural and functional/metabolic remodeling. Glial acidic fibrillary protein (GFAP), a major protein constituent of astrocyte intermediate filaments (IFs), is the hallmark of the reactive astrocytes, has pleiotropic functions and is significantly upregulated in the spinal cord after nerve injury. Here, we investigated the specific role of GFAP in glial reaction and maladaptive spinal cord plasticity following sciatic nerve spared nerve injury (SNI) in GFAP KO and wild-type (WT) animals. We evaluated the neuropathic behavior (thermal hyperalgesia, allodynia) and the expression of glial (vimentin, Iba1) and glutamate/GABA system markers (GLAST, GLT1, EAAC1, vGLUT, vGAT, GAD) in lumbar spinal cord sections of KO/WT animals. SNI induced neuropathic behavior in both GFAP KO and WT mice, paralleled by intense microglial reaction (Iba1 expression more pronounced in KO mice), reactive astrocytosis (vimentin increase) and expression remodeling of glial/neuronal glutamate/GABA transporters. In conclusion, it is conceivable that the lack of GFAP could be detrimental to the CNS as it lacks a critical sensor for neuroinflammation and morpho-functional–metabolic rewiring after nerve injury. Understanding the maladaptive morpho-functional changes of glial cells could represent the first step for a new glial-based targeted approach for mechanisms of disease in the CNS.

## 1. Introduction

Glial cells play a crucial role in synaptic transmission and plasticity of the central nervous system (CNS) [1]. Among glial cells, astrocytes have emergent properties regarding CNS synaptic function and plasticity [2,3,4,5,6]: metabolic coupling, neurotrophic support and gliotransmission (i.e., the release and reuptake of glial transmitters) can be particularly impaired during maladaptive plasticity [7,8,9,10]. On the other side, microglial cells are the resident innate immune cells and participate in synaptic reshaping in health and diseases [11]. Alterations of neuroglial networks and plasticity occur in response to external perturbations or injury in the early phases of both the CNS and peripheral nervous system [10,12,13,14]. In this context, reactive gliosis occurs as the morpho-functional activation of glial cells that first respond to pathological insults [15,16]. Morphological changes of astrocytes are usually related to the increased expression of the S100 calcium-binding protein β (S100β) or glial fibrillary acidic protein (GFAP), structural proteins of the cytoskeletal intermediate filaments (IFs). Astrocytic IFs are expressed at different stages of development and include vimentin, nestin, lamins and synemins [17]. GFAP co-polymerizes with vimentin and nestin, and its expression increases from embryonal to adult developmental stages, in contrast with other IFs. GFAP overexpression is particularly involved in pathological conditions and represents a biomarker and a prognostic factor in several CNS diseases [18,19,20,21,22]. Our group has demonstrated that modulation of reactive astrocytosis and microglial activation by several compounds including native nerve growth factor (NGF) or synthetical (BB14^®^) neurotrophins [7,23,24], inhibitors of the purinergic system (oxidized ATP, oxATP) [9] and metalloproteinases (neuroserpin and GM6001) reduced maladaptive plasticity in the CNS after peripheral nerve injury and chronic neuropathic pain [23,24,25]. However, GFAP’s role in the functional activation of astrocytes and the clinical relevance of the protein have not been fully elucidated, aside from its role in Alexander’s disease [26,27].

The radial glia of Bergmann and Müller and some cortical astrocytes express high levels of vimentin together with GFAP in the adult stage, suggesting also a functional role for vimentin in these specific populations of astrocytes and a role in pathological conditions [28,29]. Double knockout models for GFAP and vimentin demonstrated an impaired vesicle trafficking [26] but also modification of neuronal support genes and a higher inflammatory expression profile in an Alzheimer’s disease model, without affecting microglial proliferation or microglial activation [30]. Due to its role in synaptic plasticity in the hippocampus [31,32], the lack of GFAP induces neuronal degeneration and seizures and also suggests an impairment of the glutamate/GABA homeostasis. However, how GFAP modifies neurotransmitter levels and synaptic homeostasis seems to be dependent on CNS regionality as non-cell-autonomous mechanisms [13]. Given the importance of GFAP in adaptive and maladaptive plasticity of CNS, we aimed to evaluate the morpho-molecular changes in the dorsal horn of the spinal cord after spared nerve injury (SNI) of the sciatic nerve in wild-type (WT) and GFAP knockout (KO) mice.

## 2. Materials and Methods

### 2.1. Animals and SNI Model

We used B6129SF2/J mice (JAX stock #101045) as control/wild-type (WT) animals and mice carrying the homozygous GFAP^tm1Me^ mutation (JAX stock #002642) as GFAP KO animals. Animals (*n* = 36, 18 KO, 18 WT) were purchased from Jackson Laboratory (Bar Harbor, ME, USA) and maintained on a 12/12 h light/dark cycle, allowing free access to food and water. SNI of the sciatic nerve was performed on 9 animals for each group, according to the methods previously described [7,33].

Briefly, animals were anesthetized with intramuscular (i.m.) injection of ketamine (100 mg/kg) and medetomidine (0.25 mg/kg), the right sciatic nerve was exposed, and the tibial and common peroneal nerves were ligated and axotomized, leaving the sural nerve intact. In sham-operated control animals (CTR), 9 for each group, the sciatic nerve was exposed but not truncated. The animals were accordingly divided into 4 groups: (1) CTR WT, (2) CTR KO, (3) SNI WT and (4) SNI KO animals.

All surgery and experimental procedures were approved by the Ethics Committee of the University of Campania “Luigi Vanvitelli” (auth. 153/2018-PR). Animal care was in compliance with the Italian and European guidelines for the use and care of laboratory animals (EU Directive 2010/63).

### 2.2. Behavioral Testing

All behavioral tests were performed by a blinded observer. Animals were tested on day 0 (before SNI), day 7 (7 days after SNI) and day 14 (14 days after SNI) to evaluate mechanical allodynia and thermal hyperalgesia. After that, all animals were sacrificed. Mechanical allodynia was assessed using the von Frey filament test (Ugo Basile) [34,35]. Filaments were applied under the plantar surface of the right paw in the sural region, in either ascending or descending strength. The threshold was set as the lowest force that evoked a consistent, brisk withdrawal response. The time of response to a progressive force applied to the hind paw limb was assessed six times, with an interval of 5 min between stimulations. Nociceptive thresholds to radiant heat (infrared) were quantified using the plantar test apparatus (Ugo Basile) [36]. The heat source was positioned under the plantar surface of the right hind paw and activated for 20 s (cut-off) to prevent tissue damage, at a setting intensity of 7.0 to record the response latency for paw withdrawal. The injured hind limb was tested twice at each time point, with an interval of 5 min between stimulations.

### 2.3. Tissue Preparation

Fourteen days after the SNI or sham operation, all mice were sacrificed by deep anesthesia with i.m. injection of ketamine (300 mg/kg) and medetomidine (0.8 mg/kg) and perfused transcardially with saline solution (Tris HCl 0.1M/EDTA 10 mM). Spinal cords for Western blotting (WB) were removed and frozen on the day of collection. Tissues collected for immunohistochemistry (IHC) were fixed with 4% paraformaldehyde in 0.01 M phosphate-buffered saline (PBS), pH 7.4, at 4 °C before extraction. Spinal cords were removed and post-fixed in the same fixative; then, they were soaked in 30% sucrose phosphate-buffered saline (PBS) and frozen in chilled isopentane on dry ice.

### 2.4. Immunohistochemistry

Lumbar spinal cord slices 25 μm in thickness were cut on the slide microtome and processed for IHC. The following primary antibodies were used: mouse antibodies against glial fibrillary acidic protein (GFAP) (1:400; Sigma-Aldrich, Milano, Italy); rabbit antibodies against ionized calcium-binding adaptor molecule 1 (Iba1) (1:500; Wako Chemicals, Richmond, VA, USA); guinea pig antibodies against glutamate transporter (GLT1) (1:200; Chemicon Inc., Temecula, CA, USA); goat antibodies against glutamate/aspartate transporter (GLAST) (1:1000; Chemicon Inc., Temecula, CA, USA); guinea pig antibodies against vesicular glutamate transporter 1 (vGLUT1) (1:5000; Chemicon Inc., Temecula, CA, USA); mouse antibodies against vesicular GABA transporter (vGAT) (1:500; Synaptic Systems, Gottingen, Germany); rabbit antibodies against glutamic acid decarboxylase 65/67 (GAD65/67) (1:1000; Sigma-Aldrich, Milan, Italy). Briefly, sections were blocked in blocking solution (10% serum, in 0.01 M PBS/0.25% Triton-X100) for 1 h at room temperature (RT).

Each primary antibody was diluted in the blocking solution and incubated with free-floating sections for 48 h at 4 °C. After being washed in cold PBS, slices were incubated with the appropriate biotinylated secondary antibody (Vector Labs Inc., Burlingame, CA, USA; 1:200) for 90 min at RT. Samples were then processed using the Vectastain avidin–biotin peroxidase kit (Vector Labs Inc., Burlingame, CA, USA) for 90 min at RT [37], washed in 0.05 M Tris-HCl and reacted with 3.3-diaminobenzidine tetrahydrochloride (DAB; Sigma, 0.5 mg/mL in Tris-HCl) and 0.01% hydrogen peroxide. Sections were mounted on chrome alum–gelatin-coated slides, dehydrated and coverslipped.

Immunofluorescence staining was performed as previously described [38]. Tissue slices were incubated with the primary antibody (vGLUT1, vGAT and GAD65/67) for 48 h at 4 °C. Following incubation with primary antibodies, sections were incubated with the appropriate secondary antibody (Alexa Fluor 488 anti-guinea pig IgG, Alexa Fluor 488 anti-mouse IgG, Alexa Fluor 546 anti-rabbit IgG and Alexa Fluor 488 anti-rabbit IgG) (1:200; Invitrogen, Carlsbad, CA, USA) for 2 h. Sections were mounted and coverslipped with the antifading Vectashield (Vector Labs Inc., Burlingame, CA, USA).

### 2.5. Western Blotting

Perfused, fresh spinal cord tissues were homogenized in 50 mM Hepes pH 7.5, 100% glycerol, 10 mM NaCl, 10 mM dithiothreitol, 1% SDS, 5 mM EDTA and protease inhibitors (Sigma Aldrich, Milan, Italy). Samples were loaded on a 0.75 mm SDS polyacrylamide minigel (10%, 12%) that was electrophoresed at 150 V for 90 min. The proteins were transferred to nitrocellulose membranes overnight at 30 V and 4 °C. After blocking of non-specific sites with 5% milk, 20 mM Tris HCl (pH 7.4) and 0.2% Tween 20 (TBST), membranes were incubated overnight with primary antibody anti-EAAC1 (1:400), anti-vimentin (1:250) and β-actin (1:2000). After being washed in TBST, membranes were incubated with the appropriate biotinylated secondary antibody (Vector Labs Inc., Burlingame, CA, USA, 1:200) in blocking solution for 60 min at room temperature. Subsequently, they were washed in TBS and processed using the Vectastain avidin–biotin peroxidase kit (Vector Labs Inc., Burlingame, CA, USA) for 30 min at RT. Bands were revealed by reacting with 3,3-diaminobenzidine tetrahydrochloride (DAB) (Sigma Aldrich, Milan, Italy) 0.5 mg/mL Tris-HCl and 0.01% hydrogen peroxide.

The density of each band was measured with a computer-assisted imaging analysis system (MCID 7.1; Imaging Res. Inc., Ontario, CA, USA) and normalized with the corresponding β-actin band used as the internal loading control.

### 2.6. Standard Light and Confocal Microscopy

Slides were imaged with a Zeiss Axioskope 2 light microscope equipped with a high-resolution digital camera (C4742-95, Hamamatsu Photonics, Arese, Milan, Italy) for standard light microscopy. Zeiss LSM 510 Meta laser scanning microscope (Oberkochen, Germany) confocal images of dorsal horns of the lumbar spinal cord were captured at a resolution of 512 × 512 pixels. Argon laser fluorescence was used for visualization of the vGLUT1 and vGAT (excitation wavelength of 488 nm and bandpass emission filter of 505–530 nm), and HeNe laser fluorescence was used for the GAD65/67 signal (excitation wavelength of 546 nm and long-pass emission filter of 560 nm).

### 2.7. Measurements and Statistical Analysis

The expression of the IHC and WB markers in the dorsal horn of spinal cords was quantified by using the computer-assisted image analysis system MCID 7.1 (Imaging Res. Inc., Ontario, CA, USA). The microglial marker Iba1 was analyzed through a morphometric approach and expressed as a proportional area (number of positive elements relative to the scanned area). GLT1 and GLAST expression was quantified as a densitometric measurement (total density within the target outline multiplied by its area). The analysis of confocal images for vGLUT1, vGAT and GAD65/67 was also performed using a densitometric approach. Averages were obtained from five randomly selected spinal cord sections for each animal. All the data were analyzed using the Sigma-Plot 10.0 program (SPSS Erkrath, Germany). Statistical comparisons between the experimental groups were tested using the one-way ANOVA, followed by the Holm–Sidak method for all pairwise multiple comparisons (* *p* ≤ 0,05; ** *p* ≤ 0.01; *** *p* ≤0.001). Data from the molecular analysis are presented as a vertical bar chart with the mean value ± standard error (mean ± SEM), and individual data points represent the distribution of the measurements for each experimental group. Data from the behavioral analysis are presented as the mean ± SEM. Images were assembled and then the same adjustments were made for brightness, contrast and sharpness using Adobe Photoshop (Adobe Systems, San Jose, CA, USA).

## 3. Results

### 3.1. Behavioral Analysis

In WT animals, the mean baselines of mechanical and thermal thresholds before SNI (day 0) were 28.6 ± 0.5 g and 16.2 ± 0.6 s, respectively. In CTR animals, these values did not change on day 7 (28.8 ± 0.4 g and 16.6 ± 0.4 s, respectively) or day 14 (28.6 ± 0.3 g and 15.4 ± 0.4 s, respectively) (Figure 1A,B). SNI induced a neuropathic behavior on day 7, as showed by the significant reduction in the mechanical threshold in SNI animals (10.4 ± 0.5 g; *p* ≤ 0.001) compared to the sham procedures (28.1 ± 0.9 g), indicative of an allodynic state. This condition was still evident on day 14 (10.6 ± 0.6 g; *p* ≤ 0.001) (Figure 1A). The Hargreaves test confirmed the onset of a hyperalgesic state on days 7 (6.5 ± 0.6 s; *p* ≤ 0.001) and 14 (6.6 ± 0.6 s), relative to the sham (16.0 ± 0.9 s) (*p* ≤ 0.001) (Figure 1B).

In KO animals, the mean baselines of mechanical and thermal thresholds before SNI (day 0) were 27.7 ± 0.7 g and 15.4 ± 0.7 s, respectively (Figure 1A,B). SNI induced mechanical allodynia on day 7 (12.4 ± 0.6 g) and day 14 (13.4 ± 0.5 g) compared to KO-CTR (27.5 ± 0.9 g) (*p* ≤ 0.001) (Figure 1A) and a hyperalgesic state as represented by the reduction in the thermal threshold after nerve injury on day 7 (6.4 ± 0.5 s) and day 14 (7.5 ± 0.6 s), compared to the corresponding CTR value (15.4 ± 0.8 s) (*p* ≤ 0.001) (Figure 1B).

### 3.2. Reactive Gliosis Induction in the Lumbar Spinal Cord Following SNI

We measured reactive gliosis markers in both WT and KO animals 14 days after SNI to investigate the role of GFAP in the spinal neuroglial network rearrangement after nerve injury. IHC analysis showed that CTR-WT animals had a lower expression of the microglial marker Iba1 (67.7 ± 5.3) compared to CTR-KO animals (96.5 ± 9.1) (*p* ≤ 0.01) (Figure 2). Microglial activation was a common feature after SNI in both WT and KO mice, as demonstrated by the increased level of Iba1 (SNI-WT 105.6 ± 7.7; SNI-KO 132.4 ± 2.3), compared to the CTR (*p* ≤ 0.001). Iba1 expression in SNI-KO animals was higher when compared to the WT group after the same procedure (*p* ≤ 0.01) (Figure 2). No morphological changes of microglial cells were detected in both groups.

Astrocytosis was evaluated for CTR-WT and SNI-WT animals by using IHC for GFAP and WB analysis for vimentin levels. In WT mice, SNI induced a marked overexpression of GFAP (144.3 ± 17.7) and vimentin (2.3 ± 0.2), compared to CTR animals (GFAP: 91.3 ± 6.2; vimentin: 0.83 ± 0.4) (** *p* ≤ 0.001) (Figure 3). Similarly, in KO animals, molecular analysis of WB revealed an increment in vimentin expression (2.5 ± 0.2) compared to CTR-KO animals (0.55 ± 0.09; *p* ≤ 0.001) (Figure 3B,C).

Our results show that the lack of GFAP does not affect the astrocytic activation, as demonstrated by the marked increase in vimentin protein in the SNI-KO group. However, these data suggest that (1) the microglial enhancement of Iba1 expression in the dorsal horn of the spinal cord may be due to the GFAP downregulation as a sensed change in the milieu of CTR-KO animals and (2) microglial reactivity following SNI is more pronounced in GFAP-KO conditions.

### 3.3. Remodeling of Glial/Neuronal Glutamate/GABA Transporters in GFAP-KO Animals Following SNI

The role of astrocytes in synaptic homeostasis and in glutamate reuptake is well documented. Considering the importance of GABA/glutamate balance in the plastic rewiring of spinal circuitry [39], we evaluated the putative role of GFAP in the remodeling of glial and neuronal glutamate and GABA transporters in WT and KO animals after SNI.

IHC analyses revealed a reduction in GLT1 and GLAST expression following SNI in WT animals (83.6 ± 6.2 and 87.9 ± 8.8, respectively), compared to CTR levels (119.9 ± 20.9 and 110.5 ± 14.3, respectively) (** *p* ≤ 0.001; * *p* ≤ 0.01) (Figure 4).

In GFAP-KO animals, GLT1 expression was reduced after nerve injury (36.9 ± 22.1) compared to CTR-KO animals (76.6 ± 14.9) (*p* ≤ 0.001). Interestingly, GLT 1 expression was lower in CTR-KO animals and after SNI, compared to WT animals (** *p* ≤ 0.001; * *p* ≤ 0.01) (Figure 4).

Similarly, GLAST levels were lower in CTR-KO animals (69.5 ± 21.4) compared to WT mice (110.5 ± 14.3) (*p* ≤ 0.001). However, GLAST levels were increased in KO mice following SNI (88.1 ± 13.1) compared to CTR (69.5 ± 21.4) (*p* ≤ 0.01) (Figure 4).

We measured EAAC1 expression through WB analysis to evaluate the neuronal contribution to glutamate homeostasis. We found a similar increase in the neuronal glutamate transporter EAAC1 in both WT and GFAP-KO animals after SNI (WT: 2.69 ± 0.3; KO: 2.73 ± 0.2) compared to the values in CTR animals (WT: 0.9 ± 0.2; KO: 0.83 ± 0.1) (*p* ≤ 0.001) (Figure 4).

These data demonstrated that the expression of the glial glutamate transporters GLT1 and GLAST is related to GFAP absence and is differentially modulated by SNI.

To further characterize the role of GFAP in synaptic transmission homeostasis, we evaluated the expression of the vesicular transporters for GABA and glutamate (vGAT and vGLUT) and the levels of GAD65/67, the two isoforms of the decarboxylase that enzymatically synthesizes GABA from glutamate.

No significant differences in vGLUT expression were found in WT animals after sham procedures (133.1 ± 13.6) or SNI (138.3 ± 13.4). In contrast, in GFAP-KO animals, vGLUT expression increased after SNI (197.8 ± 7.8) and was significantly higher compared to both KO-CTR (134.2 ± 10.7) and WT-SNI values (*p* ≤ 0.001) (Figure 5). However, the CTR levels of vGLUT did not differ between KO and WT animals.

Analysis of vGAT expression revealed a significant increase after SNI (WT: 188.8 ± 12.4, KO: 174.1 ± 18.4) compared to CTR animals (WT: 112.3 ± 13.7, KO: 119.4 ± 17.3) (*p* ≤ 0.001) (Figure 5). No difference was detected when comparing the WT and KO animals.

GAD65/67 densitometric values increased after SNI (189.7 ± 7.4) in WT animals when compared to CTR mice (116.7 ± 13.2) (*p* ≤ 0.001). Similarly, the values of GAD65/67 detected in KO animals after SNI (179.6 ± 12.2) were higher than the CTR values (146.7 ± 12.2; *p* ≤ 0.01) (Figure 5). GAD65/67 levels in WT-CTR animals were also lower than those in KO-CTR mice (*p* ≤ 0.01).

These data demonstrated that GFAP KO animals use different mechanisms to balance glutamate/GABA homeostasis in the dorsal horn of the spinal cord, at baseline and following SNI.

## 4. Discussion

Our work has investigated the specific role of the GFAP in the maladaptive synaptic process following peripheral nerve injury.

We demonstrated a peculiar morpho-molecular phenotype in GFAP null mice, even in the intact spinal cord circuitry. We have found a higher density of the microglial marker Iba1 in the dorsal horn of the spinal cord in CTR-KO mice compared to WT animals that, to the best of our knowledge, has never been reported before. The role of the increased microglial surveillance is not known; however, evidence has shown that GFAP null mice are less resistant to spinal cord injuries [40] and cerebral ischemia [41]. The microglial support of synaptic plasticity is one of the emergent and interesting properties of these cells [42]. The increased expression of Iba1 positive elements after nerve injury could depend on GFAP influencing embryological development. For instance, microglia are the only resident CNS cellular element that does not originate from the neural tube [43]. How GFAP loss could be potentially responsible for the increased colonization of these cells from the yolk sac to the neural tube is unknown and opens the search for novel pathways of migration and cell–cell signaling. However, the absence of GFAP and/or vimentin induced an altered inflammatory profile without affecting microglial morphology, proliferation or activation [30]. Altogether, these data suggest a regionalized GFAP influence on microglial cells, although more data are required to validate this hypothesis.

Due to the absence of one of the main astrocytic markers [16], reactive astrocytosis was evaluated for the expression of vimentin, which was increased following SNI. Due to the role of IFs in the intracellular trafficking of the vesicles containing receptors and gliotransmitters [18], we analyzed the expression of glial glutamate transporters. Our data confirm a different balance of neurotransmitter metabolism and reuptake, as we found that GFAP-KO animals had a reduced expression of the glial glutamate transporters (GLAST and GLT1) at baseline that could be directly linked to the absence of GFAP and might be responsible for impaired glutamate uptake and spillover, altered LTP and excitotoxicity [7,33,34]. Our group proved that the reduction in astrocyte-specific glutamate transporters after nerve injury is linked to activation of calpain I, a Ca^2+^-dependent protease involved in the degradation of not only cytoskeletal structural proteins (including GFAP) but also membrane astrocytic transporters, such as GLT1 [44,45,46,47]. Moreover, maladaptive rearrangement and clustering of GLAST expression after CNS injury are correlated with loco-regional reactive astrocytosis [48], suggesting that astrocytic reaction and IF expression regulate the CNS regional architecture of the astrocytic membrane proteins [26].

Recently, the regulation of the expression of many membrane proteins (including transporters and receptors) has been linked to cholesterol content in the plasma membrane [49,50]. Neuroinflammatory reaction following nerve injury is characterized by an abundance of lipid rafts and cholesterol content in microglia and astrocytes, with increased levels of inflammatory receptors (i.e., Toll-like receptor-4 (TLR4)) and evidence of protein complex assemblies in the plasma membrane. Removal of cholesterol from rafts by administration of apolipoprotein A-I binding protein (AIBP) quenched the activation of glial cells involving the inflammatory-like Toll-like receptor-4 (TLR4) and ameliorated chronic pain states [49]. Therefore, cholesterol-rich membrane rafts in glial cells serve as local organizing matrices for membrane receptors and channels involved in neuroinflammation and pain processing in the spinal cord [50]. How this mechanism links together GFAP expression and neuroinflammation following injury is far from being elucidated and further increases the complexity of CNS response after injury [13,42,51].

The changes in the glial glutamate transporter expression levels and the consequent impairment of glutamate uptake following SNI induced an increase in vGLUT levels in KO animals, as a potential compensatory mechanism to counterbalance the excessive glutamatergic tone. Our group previously demonstrated the positive effects of vGLUT overexpression in the SNI model after blockage of the purinergic receptors [9]. Similarly, the upregulation of the neuronal glutamate transporter EAAC1 in the lumbar spinal cord of KO mice could represent another compensatory mechanism to reduce glutamate function, as previously demonstrated [7,8,9,10,39].

The GABA/glutamate balance following SNI was differently modulated in KO and WT animals. In particular, the reduction in GLT1 expression, which was observed in both groups and is a well-documented maladaptive plasticity phenomenon [7,8,9], was even more pronounced in the KO animals. Interestingly, we observed an unexpected increase in GLAST levels after SNI in the KO group, suggesting that the increase in GLAST and vGLUT expression could balance the very low levels of GLT1. Moreover, the GABA vesicular transporter vGAT and the GAD65/67 levels increased after SNI in both WT and KO mice. Therefore, they do not seem to be linked to GFAP expression. The increase in the two GABAergic markers could be an insufficient adaptive change of the circuitry to the increased glutamatergic tone, as already proposed in previous experiments [1,9].

GFAP-KO mice developed neuropathic behavior after SNI with similar temporal and behavioral features to WT animals. These results highlighted the role of the glutamate-induced maladaptive remodeling of the spinal circuitry in neuropathic behavior rather than a direct connection with the lack of GFAP. Converging pathways adjusting glutamatergic tone can be detrimentally influenced by GFAP deficiency [28,32,40], but maladaptive plasticity induced by SNI might be counterbalanced by mechanisms that are independent of GFAP expression.

## 5. Conclusions

It remains unclear how GFAP could modulate microglial density, glutamatergic and GABAergic synaptic transmission, transport and metabolism. However, our study demonstrated the occurrence of a neuropathic behavior and a peculiar morpho-molecular rearrangement in GFAP null mice after nerve injury. The role of astrocytic IFs is essential to the maintenance of CNS homeostasis, as demonstrated by the higher susceptibility of IF knockout animal models to maladaptive changes after damage [28,32,40]. However, to date, only a few studies have focused on the specific role of IFs and their involvement in the maladaptive rewiring of the CNS in neuroinflammatory and neurodegenerative mechanisms. For instance, lipoxidation of the GFAP cysteine residue prevents the GFAP–vimentin co-heteropolymerization and contributes to GFAP disruption or aggregation in pathological situations associated with oxidative stress [52]. Moreover, maladaptive changes in GFAP structure and consequent astrogliosis and neurodegeneration have been linked to GFAP hyperpalmitoylation, which might represent a new target for the neurodegenerative process [53]. Therefore, it is conceivable that the lack of GFAP could be detrimental to the CNS as it lacks a critical sensor for stress and neuroinflammation.

The emerging role of glial cells and reactive gliosis as a primer for neuronal dysfunction and death is changing our view of CNS disease. A loco-regional gliopathy, therefore, could induce a selective alteration of a specific neuronal microenvironment that induces selective neuronal death. Understanding the maladaptive morpho-functional changes of glial cells could represent the first step for a new glial-based targeted approach for mechanisms of disease in the CNS.

## Figures and Tables

**Figure 1 cells-11-01224-f001:**
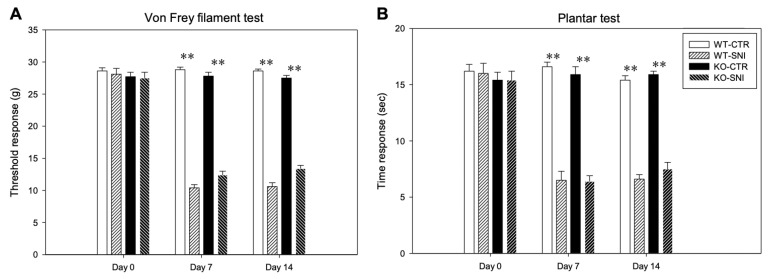
Behavioral analysis. WT and KO mice, tested with von Frey filament test (**A**) and thermal plantar test (**B**) for baseline sensitivity (Day 0) and 7 (Day 7) and 14 days (Day 14) after SNI. Data are expressed as the mean ± SEM (** *p* ≤ 0.001).

**Figure 2 cells-11-01224-f002:**
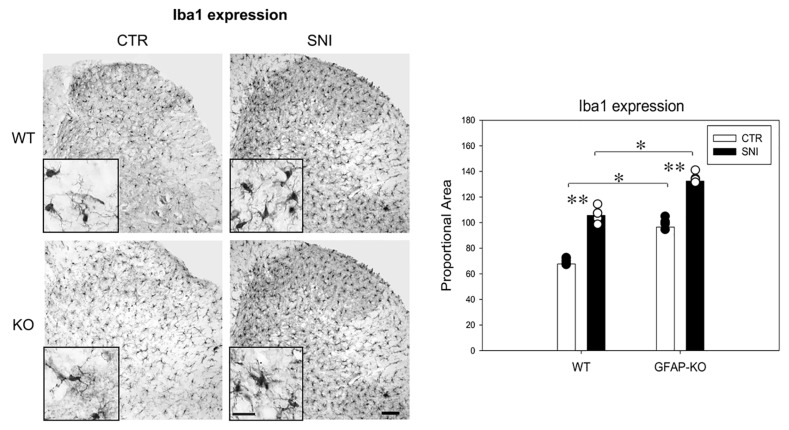
Microglial Iba1 expression in the dorsal horn of the spinal cord. Iba1 expression increased after SNI in both WT and KO animals. Iba1 level detected in CTR-KO animals was higher than that in CTR-WT mice (magnification 10×; scale bar, 50 μm). **Inset**: higher magnification (40×) view of microglial cells, showing morphological features in CTR and nerve-injured mice (scale bar, 50 μm). (Data are expressed as the mean ± SEM with individual data points; ** *p* ≤ 0.001; * *p* ≤ 0.01.).

**Figure 3 cells-11-01224-f003:**
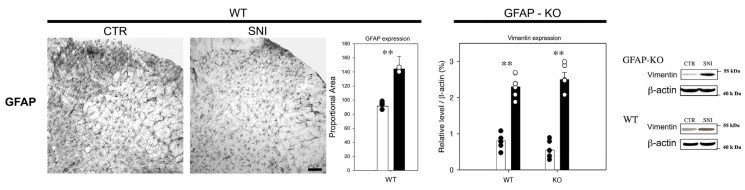
(**A**) GFAP expression in the dorsal horn of the spinal cord (magnification 10×; scale bar, 50 μm). GFAP expression significantly increased after SNI in WT mice. (**B**,**C**) WB vimentin expression in WT and GFAP-KO animals showed a marked increase after SNI compared to CTR (data are expressed as the mean ± SEM. ** *p* ≤ 0.001).

**Figure 4 cells-11-01224-f004:**
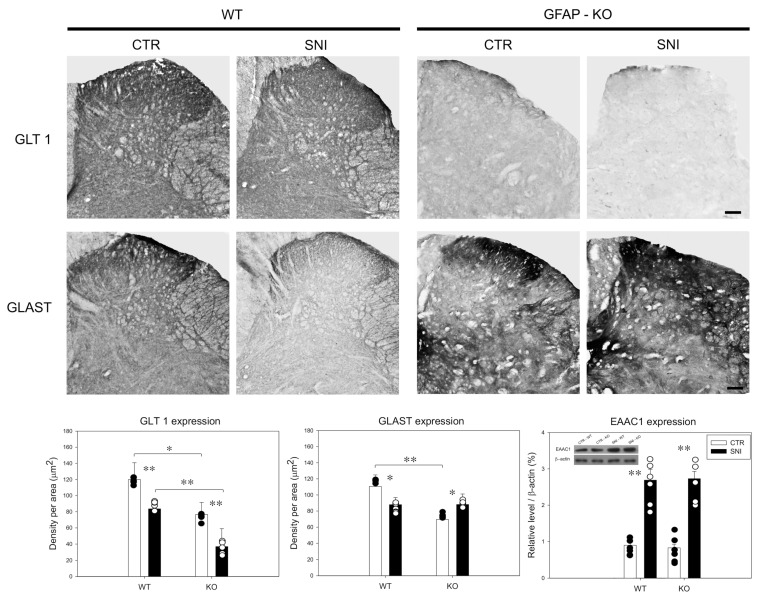
Expression of glial (GLT 1, GLAST) and neuronal (EAAC1) glutamate transporters in the dorsal horn of the spinal cord. Data are expressed as the mean ± SEM (** *p* ≤ 0.001; * *p* ≤ 0.01).

**Figure 5 cells-11-01224-f005:**
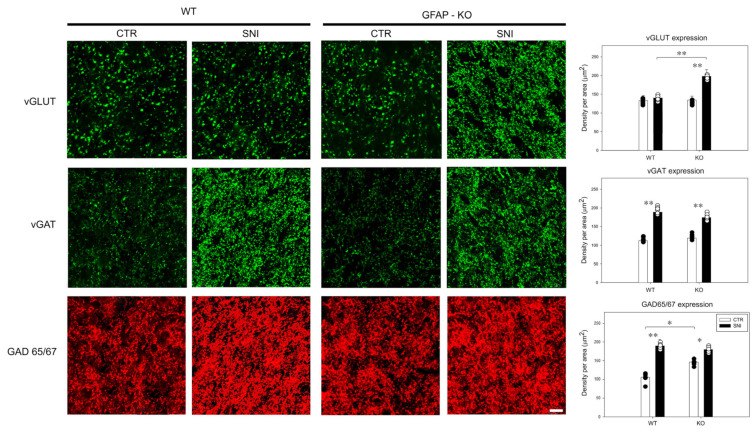
Confocal images of dorsal horns of the lumbar spinal cord in WT (magnification 100×; scale bar, 10 µm) and GFAP-KO mice (magnification 20×; scale bar, 50 µm) in CTR and SNI mice, immunostained for vGLUT, vGAT and GAD. Data are expressed as the mean ± SEM (** *p* ≤ 0.001; * *p* ≤ 0.01).

## Data Availability

The datasets and materials generated and analyzed during the current study are available from the corresponding author upon reasonable request.

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
