# Peer review of "Altered Spinal Homeostasis and Maladaptive Plasticity in GFAP Null Mice Following Peripheral Nerve Injury"

_cells, 2022, doi:10.3390/cells11071224_

Round 1

Reviewer 1 Report

The authors assess the role of GFAP in the response to spared nerve injury model in mice. They show that absence of GFAP, the neuropathic behaviour response is comparable to WT mice while the expression pattern of glutamate and GABA systemic markers are altered. The study design is sound and the results interesting, however there are some issues the authors need to deal with:

Point 1. Introduction, line 45: Please rephrase ”IFs are a large group of astrocytic cytoskeletal proteins“, since IFs are not a group of proteins, they are filaments composed of proteins.

Point 2. Introduction, line 57: “Double knockout models for GFAP and vimentin demonstrated … a decrease of neuronal support genes ….[21]. Reference 21 does not show decrease of neuronal support genes, it shows a slight increase in neuronal support genes in mice lacking GFAP and vimentin in an Alzheimer’s disease model. The authors need to rephase this sentence.

Point 3. Results: line 218. “ These data suggest that 1) microglial expression in the dorsal horn spinal cord of CTR animals is linked to GFAP expression”  This statement is unclear to me. Microglial expression of what? How is that expression linked to GFAP expression? The authors need to rephrase.

Point 4. Results, Line 220. “Lack of GFAP, moreover, has no effect on the astrocytic activation and reactive astrocytosis as expressed by the marked increase of vimentin expression.“

The authors show an increase in vimentin expression in GV mice following SNI. From this results it can not be concluded that GFAP deletion has NO effect on astrocytic activation without assessing the same response  to SNI in the WT mice. The data shows that astrocyte activation as measured by vimentin expression does occur but not whether the response is at a comparable level to WT mice after SNI.

The authors should include Western blot analysis of vimentin expression in WT control and WT after SNI in the data shown in figure 2B.

Point 5. Results, Fig. 4: Why do the authors show different magnification of images from WT (100x) and from GFAP KO (20x) spinal cords? This difference in magnification does not make sense and makes it impossible for the reader to visually compare the images.

The authors should provide images with the same magnification for WT and GFAP KO (100x is preferred over 20x).

Point 6. Discussion, Line 320: “However, absence of GFAP and/or vimentin induced a higher inflammatory profile without affecting microglial proliferation or activation [21]”.

I suggest to change "higher" to "altered".

Author Response

The authors assess the role of GFAP in the response to spared nerve injury model in mice. They show that absence of GFAP, the neuropathic behaviour response is comparable to WT mice while the expression pattern of glutamate and GABA systemic markers are altered. The study design is sound and the results interesting, however there are some issues the authors need to deal with:

Point 1. Introduction, line 45: Please rephrase ”IFs are a large group of astrocytic cytoskeletal proteins“, since IFs are not a group of proteins, they are filaments composed of proteins.

R: We rephrased the sentence with the following “ Morphological changes of astrocytes are usually related to the increased expression of the S100 calcium-binding protein β (S100β) or glial fibrillary acidic protein (GFAP), structural proteins of the cytoskeletal intermediate filaments (IFs). Astrocytic IFs are expressed at different stages of development and include vimentin, nestin, lamins, and synemins” (please see lines 42-46, Introduction).

Point 2. Introduction, line 57: “Double knockout models for GFAP and vimentin demonstrated … a decrease of neuronal support genes ….[21]. Reference 21 does not show decrease of neuronal support genes, it shows a slight increase in neuronal support genes in mice lacking GFAP and vimentin in an Alzheimer’s disease model. The authors need to rephase this sentence.

R: We corrected the sentence: “Double knockout models for GFAP and vimentin demonstrated an impaired vesicle trafficking [17] but also modification of neuronal support genes and a higher inflammatory expression profile in an Alzheimer’s disease model, without affecting microglial proliferation or microglial activation” (please see lines 56-59).

Point 3. Results: line 218. “ These data suggest that 1) microglial expression in the dorsal horn spinal cord of CTR animals is linked to GFAP expression”  This statement is unclear to me. Microglial expression of what? How is that expression linked to GFAP expression? The authors need to rephrase.

R: We thank the reviewer for the relevant comment. Microglia constitute the first line of defense in the CNS and rapidly sense any modification in the surrounding milieu. Moreover, microglia and astrocytes are involved in complex crosstalk in physiological conditions and disease (Matejuk and Ransohoff, doi.org/10.3389/fimmu.2020.01416). Therefore, GFAP and/or vimentin deficiency alters gene expression in both astrocytes and microglia in wild-type mice (Kamphuis V et al, 2015, DOI: 10.1002/glia.22800). Our result about the increase of Iba1 expression in CTR-KO animals compared to CTR-WT matchs with the aforementioned evidence and can be due to the modifications of the microenvironment sensed by microglial cells.

We rephrased with the following sentences: Our results show that the lack of GFAP does not affect the astrocytic activation, as demonstrated by the marked increase of the vimentin protein in the SNI-KO group. However, these data suggest that 1) the microglial enhancement of Iba1 expression in the dorsal horn of the spinal cord may be due to the GFAP downregulation as a sensed change in the milieu of CTR-KO animals and 2) microglial reactivity following SNI is more pronounced in GFAP-KO conditions. (please see lines 217-222).

Point 4. Results, Line 220. “Lack of GFAP, moreover, has no effect on the astrocytic activation and reactive astrocytosis as expressed by the marked increase of vimentin expression.“ The authors show an increase in vimentin expression in GV mice following SNI. From this results it can not be concluded that GFAP deletion has NO effect on astrocytic activation without assessing the same response  to SNI in the WT mice. The data shows that astrocyte activation as measured by vimentin expression does occur but not whether the response is at a comparable level to WT mice after SNI. The authors should include Western blot analysis of vimentin expression in WT control and WT after SNI in the data shown in figure 2B.

R: We thank the reviewer for the relevant comment. We included the Western blot analysis of vimentin expression in WT control and WT-SNI in Figure 2.

Point 5. Results, Fig. 4: Why do the authors show different magnification of images from WT (100x) and from GFAP KO (20x) spinal cords? This difference in magnification does not make sense and makes it impossible for the reader to visually compare the images. The authors should provide images with the same magnification for WT and GFAP KO (100x is preferred over 20x).

R: We improved the Fig 4 by showing images at the same magnification (100x).

Point 6. Discussion, Line 320: “However, absence of GFAP and/or vimentin induced a higher inflammatory profile without affecting microglial proliferation or activation [21]”. I suggest to change "higher" to "altered".

R: We appreciated the suggestion. We rephrased the sentence as suggested by the reviewer (please see line 320).

Reviewer 2 Report

The activation of glial cells in the central nervous system (CNS) is a leading cause of the maladaptive response following nerve injury. Astrocytes experience altered transcriptional regulation and undergo biochemical, morphological, metabolic, and physiological changes in this process, which is referred to as reactive gliosis. An extremely essential component of astrocyte intermediate filaments is a glial acidic fibrillary protein (GFAP), which is strongly upregulated as a response to injury in the spinal cord.    Authors aimed to examine the specific role of GFAP in astrocytic reaction and maladaptive spinal plasticity after spared nerve injury (SNI) of the sciatic nerve, because of GFAP's functions, ranging from physiological plasticity to reactive response to pathological stimuli. Therefore, authors analysed GFAP-KO mice lumbar spinal cord sections were examined for glial markers (vimentin, Iba1) and glutamate/GABA markers (GLAST, GLT1, EAAC1, vGLUT, vGAT, GAD). In comparison with wildtype (WT) mice, the authors demonstrated that GFAP deletion induces spinal homeostasis alteration in control mice and neuropathic behaviour in mice following SNI. Overall I found the article informative to read. However, I have the following concerns,  
  1. The model, GFAP KO might not be an ideal animal model to study the influence of GFAP on nerve injury due to the knockdown of GFAP from birth. Further, the cellular system is already adapted to the absence of GFAP. Therefore, a study on inducible Cre based GFAP flox mice would provide insights into the role of GFAP in nerve injury.
  2. What is the morphology of astrocytes and microglia under different conditions, which authors could show through sholl analysis? Since GFAP is a cytoskeleton marker, it is critical to show cellular morphology.
  3. How does the CNS injury change in the absence of GFAP?
  4. Authors should present data representing individual data points. 
  5. Authors should pay attention to several language errors and usage. 

Author Response

To Prof. Dr. Naweed I. Syed

Section Editor-in-chief

Cells

Ref: [Cells] Manuscript ID: cells-1611752 – revision

Dear Editor and Reviewer #2

we appreciate the prompt and constructive review that our paper has received. We are now sending a revised version of the manuscript, having addressed all reviewers comments. We hope that the responses and the revised manuscript are satisfactory, and meet with your approval.

Reviewer#2

The activation of glial cells in the central nervous system (CNS) is a leading cause of the maladaptive response following nerve injury. Astrocytes experience altered transcriptional regulation and undergo biochemical, morphological, metabolic, and physiological changes in this process, which is referred to as reactive gliosis. An extremely essential component of astrocyte intermediate filaments is a glial acidic fibrillary protein (GFAP), which is strongly upregulated as a response to injury in the spinal cord.    Authors aimed to examine the specific role of GFAP in astrocytic reaction and maladaptive spinal plasticity after spared nerve injury (SNI) of the sciatic nerve, because of GFAP's functions, ranging from physiological plasticity to reactive response to pathological stimuli. Therefore, authors analysed GFAP-KO mice lumbar spinal cord sections were examined for glial markers (vimentin, Iba1) and glutamate/GABA markers (GLAST, GLT1, EAAC1, vGLUT, vGAT, GAD). In comparison with wildtype (WT) mice, the authors demonstrated that GFAP deletion induces spinal homeostasis alteration in control mice and neuropathic behaviour in mice following SNI. Overall I found the article informative to read. However, I have the following concerns:

  1. The model, GFAP KO might not be an ideal animal model to study the influence of GFAP on nerve injury due to the knockdown of GFAP from birth. Further, the cellular system is already adapted to the absence of GFAP. Therefore, a study on inducible Cre based GFAP flox mice would provide insights into the role of GFAP in nerve injury.

R: GFAP KO mice develop normally and display no gross alterations in CNS morphology (Davila D. et al., 2023, doi: 10.3389/fncel.2013.00272). Based on this evidence, with the present work, we thought at first to study the effect of long-term GFAP absence in a variety of spinal cord molecular processes during CTR conditions and after spared nerve injury. However, we are planning to use an inducible Cre-GFAP flox approach to have a temporal control of gene recombination in astrocytes and assess the short-term effects of GFAP inactivation in the adult mice, gaining new insights in the field of CNS neuropathies induced by peripheral nerve injury, with a higher translational value.

  1. What is the morphology of astrocytes and microglia under different conditions, which authors could show through sholl analysis? Since GFAP is a cytoskeleton marker, it is critical to show cellular morphology.

R: We appreciated the relevant comment raised by the reviewer. The microscopic evaluation of the microglia morphology in the KO-CTR and KO-SNI reproduced the features of the microglia in WT-CTR and the classic activation in WT-SNI, without clear differences in terms of processes arborization and ameboid shape of the body. Therefore, we did not conduct the Sholl analysis. We have inserted inbox pictures of microglial cells in fig. 1, to highlight morphological features. Unfortunately, we did not perform any targeting of astrocytes for morphological analysis, but we will consider it for future studies. However, it has been demonstrated that eGFP labeled astrocytes do not show any clear morphological change in the area covered by a single astrocyte and in the arborization of their processes in both a model of GFAP-KO and GFAP-VIM-KO mice (Kamphuis V et al, 2015, DOI: 10.1002/glia.22800).

  1. How does the CNS injury change in the absence of GFAP?

R: Studies using GFAP null mice attributed several activities to GFAP including the suppression of neuronal proliferation and neurite extension in the mature brain, alteration of glial-scar formation and blood-brain-barrier, the regulation of blood flow following ischemia, supporting-function in myelination. However, findings for many of these roles have been variable among laboratories, pointing to the presence of unappreciated complexity in GFAP function. One complexity may be regional differences in GFAP activities; others are yet to be discovered (Brenner et al, 2015 10.1016/j.neulet.2014.01.055).

  1. Authors should present data representing individual data points. 

R: We improved the layout of the graphs (vertical bar chart representing the group mean value ± standard error with individual data points) as suggested by the reviewer.

  1. Authors should pay attention to several language errors and usage. 

R: We carefully read the manuscript and corrected language errors and usage throughout the manuscript.

Round 2

Reviewer 2 Report

The authors have not really put effort into providing any experimental data for my first-time feedback.
In general, the authors simply justify their points in a different manner rather than providing some proof.
Therefore, I would recommend the editor either reject the paper after considering the quality of the work or not meeting the journal standard or refer to brain sciences.

Author Response

We regret to read the reviewer's disappoint to our revision. We are aware of the descriptive nature of our work and of the necessity to improve and extend the experimental design. However, our group has described and explained several aspects of reactive gliosis in the CNS following nerve injury. According to the reviewers' and editor's comments, we have rephrased and made clearer our paper, justifying our sentences in the light of recent literature, hoping to meet your approval.